SCIENCE FORUM

# Consensus-based guidance for conducting and reporting multi-analyst studies

**Abstract**  Any large dataset can be analyzed in a number of ways, and it is possible that the use of different analysis strategies will lead to different results and conclusions. One way to assess whether the results obtained depend on the analysis strategy chosen is to employ multiple analysts and leave each of them free to follow their own approach. Here, we present consensus-based guidance for conducting and reporting such multi-analyst studies, and we discuss how broader adoption of the multi-analyst approach has the potential to strengthen the robustness of results and conclusions obtained from analyses of datasets in basic and applied research.

**BALAZS ACZEL\*†, BARNABAS SZASZI\*†, GUSTAV NILSONNE, OLMO R VAN DEN AKKER, CASPER J ALBERS, MARCEL ALM VAN ASSEN, JOJANNEKE A BASTIAANSEN, DANIEL BENJAMIN, UDO BOEHM, ROTEM BOTVINIK-NEZER, LAURA F BRINGMANN, NIKO A BUSCH, EMMANUEL CARUYER, ANDREA M CATALDO, NELSON COWAN, ANDREW DELIOS, NOAH NN VAN DONGEN, CHRIS DONKIN, JOHNNY B VAN DOORN, ANNA DREBER, GILLES DUTILH, GARY F EGAN, MORTON ANN GERNSBACHER, RINK HOEKSTRA, SABINE HOFFMANN, FELIX HOLZMEISTER, JUERGEN HUBER, MAGNUS JOHANNESSON, KAI J JONAS, ALEXANDER T KINDEL, MICHAEL KIRCHLER, YORAM K KUNKELS, D STEPHEN LINDSAY, JEAN-FRANCOIS MANGIN, DORA MATZKE, MARCUS R MUNAFÒ, BEN R NEWELL, BRIAN A NOSEK, RUSSELL A POLDRACK, DON VAN RAVENZWAAIJ, JÖRG RIESKAMP, MATTHEW J SALGANIK, ALEXANDRA SARAFOGLOU, TOM SCHONBERG, MARTIN SCHWEINSBERG, DAVID SHANKS, RAPHAEL SILBERZAHN, DANIEL J SIMONS, BARBARA A SPELLMAN, SAMUEL ST-JEAN, JEFFREY J STARNS, ERIC LUIS UHLMANN, JELTE WICHERTS AND ERIC-JAN WAGENMAKERS**

**\*For correspondence:**
aczel.balazs@ppk.elte.hu (BA);
szaszi.barnabas@ppk.elte.hu (BS)

†These authors contributed equally to this work

**Competing interest:** See page

## Introduction

Empirical investigations often require researchers to make a large number of decisions about how to analyze the data. However, the theories that motivate investigations rarely impose strong restrictions on how the data should be analyzed. This means that empirical results typically hinge on analytical choices made by just one or a small number of researchers, and raises the possibility that different – but equally justifiable – analytical choices could lead to different results (***Figure 1***).

This "analytical variability" may be particularly high for datasets that were not initially collected for research purposes (such as electronic health records) because data analysts might know relatively little about how those data were collected and/or generated. However, when analyzing such datasets – and when making decisions based on the results of such analyses – it is important to be aware that the results will be subject to higher levels of analytical variability than the results obtained from analyses of data from, say, clinical trials. A recent example of the perils of analytical variability is provided by two articles in the journal *Surgery* that used the same dataset to investigate the same question: does the use of a retrieval bag during laparoscopic appendectomy reduce surgical site infections?

Each paper used reasonable analysis, but there were notable differences between them in how they addressed inclusion and exclusion criteria, outcome measures, sample sizes, and covariates. As a result of these different analytical choices, the two articles reached opposite conclusions: one paper reported that using a retrieval bag reduced infections (*Fields et al., 2019*), and the other reported that it did not (*Turner et al., 2019*; see also *Childers and Maggard-Gibbons, 2021*). This and other medical examples (*de Vries et al., 2010*; *Jivanji et al., 2020*; *Shah et al., 2021*) illustrate how independent analysis of the same data can reach different, yet justifiable, conclusions.

The robustness of results and conclusions can be studied by evaluating multiple distinct analysis options simultaneously (e.g., vibration of effects [*Patel et al., 2015*] or multiverse analysis [*Steegen et al., 2016*]), or by employing a "multi-analyst approach" that involves engaging multiple analysts to independently analyze the same data. Rather than exhaustively evaluating all plausible analyses, the multi-analyst approach examines analytical choices that are deemed most appropriate by independent analysts. *Botvinik-Nezer et al., 2020a*, for example, asked 70 teams to test the same hypotheses using the same functional magnetic resonance imaging dataset. They found that no two teams followed the same data preprocessing steps or analysis strategies, which resulted in substantial variability in the teams' conclusions. This and other work (*Bastiaansen et al., 2020*; *van Dongen et al., 2019*; *Salganik et al., 2020*; *Silberzahn et al., 2018*; *Dutilh et al., 2018*; *Fillard et al., 2011*; *Starns et al., 2019*; *Maier-Hein et al., 2017*; *Poline et al., 2006*) confirms how results can depend on analytic choices.

Although the multi-analyst approach will be new to many researchers, it has been in use since the 19th century. In 1857, for example, the Royal Asian Society asked four scholars to independently translate a previously unseen inscription to verify that the ancient Assyrian language had been deciphered correctly. The almost perfect overlap between the solutions indicated that "they have Truth for their basis" (*Fox Talbot et al., 1861*). The same approach can be used to analyze data today. With just a few co-analysts, the multi-analyst approach can be informative about the analytic robustness of results and conclusions. When the results of independent data analyses converge, more confidence in the conclusions is warranted. However, when the results diverge, confidence will be reduced,

and scientists can examine the reasons for these discrepancies and identify potentially meaningful moderators of the results. With enough co-analysts, it is possible to estimate the variability among analysis strategies and attempt to identify factors explaining this variability.

The multi-analyst approach is still rarely used, but we argue that many disciplines could benefit from its broader adoption. To help researchers overcome practical challenges, we provide consensus-based guidance (including a checklist) to help researchers surmount the practical challenges of preparing, conducting, and reporting multi-analyst studies.

## Methods

To develop this guidance, we recruited a panel of 50 methodology experts who followed a preregistered 'reactive-Delphi' expert consensus procedure (*McKenna, 1994*). We adopted this procedure to ensure that the resulting guidance represents the shared thinking of relevant experts and that it incorporates their topic-related insights. The applied consensus procedure and its reporting satisfy the recommendations of CREDES (*Jünger et al., 2017*), a guidance on conducting and reporting Delphi studies. A flowchart of the Delphi expert consensus procedure is available at https://osf.io/pzkcs/.

### *Preparation*
Preregistering the project
Before the start of the project, on 11 November 2020, a research plan was compiled and uploaded to a time-stamped repository at https://osf.io/dgrua. During the project, we followed the preregistered plan in all respects except implementing slight changes in the wording of the survey questions to improve comprehension and not using R to analyze our results. We declared that we would share the R code and codebook of our analyses, but the project ultimately did not require us to conduct analyses in R. Instead, we shared our code in Excel and ODS format at https://osf.io/h36qy/.

Creating the initial multi-analyst guidance draft
Before the expert consensus process, the first three authors and the last author (henceforth: proposers) created an initial multi-analyst guidance draft after brainstorming and reviewing all the previously published multi-analyst-type projects they were aware of *Bastiaansen et al.,*

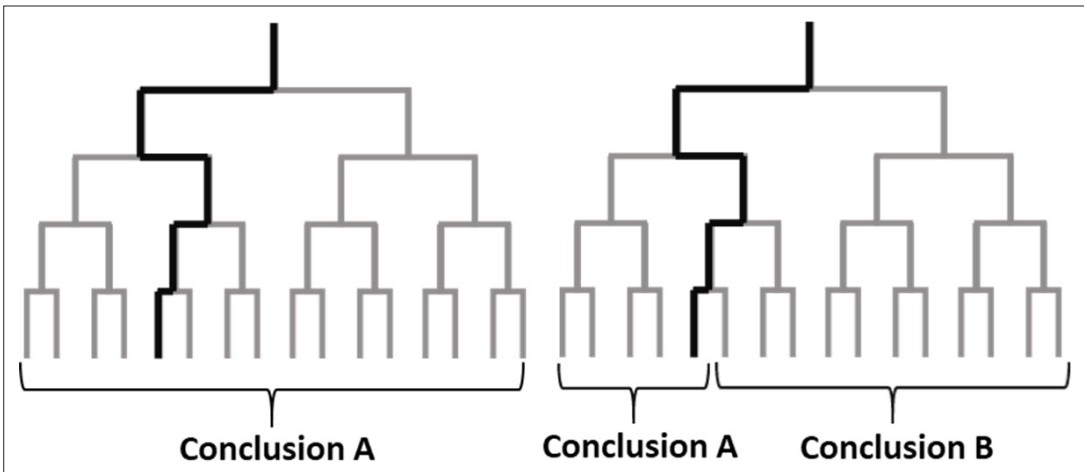

**Figure 1.** Analysis choices and alternative plausible paths. The analysis of a large dataset can involve a sequence of analysis choices, as depicted in these schematic diagrams. The analyst first must decide between two options at the start of the analysis (top), and must make three additional decisions during the analysis: this leads to 16 possible paths for the analysis (grey lines). The left panel shows an example in which all possible paths lead to the same conclusion; the right panel shows an example in which some paths lead to conclusion A and other paths lead to conclusion B. Unless we can test alternative paths, we cannot know if the results obtained by following one particular path (thick black line) are robust, or if other plausible paths would lead to different results.

*2020*; *van Dongen et al., 2019*; *Salganik et al., 2020*; *Silberzahn et al., 2018*; *Botvinik-Nezer et al., 2020a*; *Dutilh et al., 2018*; *Fillard et al., 2011*; *Starns et al., 2019*; *Maier-Hein et al., 2017*; *Poline et al., 2006*. This initial document is available here: https://osf.io/kv8jt/.

Recruiting experts

The proposers contacted 81 experts to join the project. The contacted experts included all the organizers of previous multi-analyst projects known at the time (*Bastiaansen et al., 2020*; *van Dongen et al., 2019*; *Salganik et al., 2020*; *Silberzahn et al., 2018*; *Botvinik-Nezer et al., 2020a*; *Dutilh et al., 2018*; *Fillard et al., 2011*; *Starns et al., 2019*; *Maier-Hein et al., 2017*; *Poline et al., 2006*), as well as the members of the expert panel from another methodological consensus project (*Aczel et al., 2020*). The previous projects were identified by conducting an unsystematic literature search and by surveying researchers in social media. Of the 81 experts, 3 declined our invitation and 50 accepted the invitation and participated in the expert consensus procedure (their names are available at https://osf.io/fwqvp/), while 28 experts did not respond to our call.

*Preparatory rounds*

Upon joining the project, the experts received a link to the preparatory online survey (available at https://osf.io/kv8jt/) which included the initial

Multi-Analyst Guidance draft where they had the option to comment on each of the items and the overall content of the guidance.

Based on the feedback received from the preparatory online survey, the proposers updated and revised the initial Multi-Analyst Guidance. This updated document was uploaded to an online shared document and was sent out to the experts who had the option to edit and comment on the content. Again, based on feedback, the proposers revised the content of the document, and this new version was included in the expert consensus survey.

*Consensus survey*

The expert consensus questionnaire was sent out individually to each expert first on 8 February 2021 in the following Qualtrics survey available at https://osf.io/wrpnq/. The consensus survey approach had the advantage of minimizing potential biases in the experts' judgments: the questions were posed in a neutral way, experts all received the same questions, and experts did not see the responses of the other experts or any reaction of the project organizers. The survey contained the ten recommended practices grouped into the following five stages:( i) recruiting co-analysts; (ii) providing the dataset, research questions, and research tasks; (iii) conducting the independent analyses; (iv) processing the results; (v) reporting the methods and results. The respondents were asked to rate each of the ten recommended

practices on a nine-point Likert-type scale ('I agree with the content and wording of this guidance section' ranging from "1-Disagree" to "9-Agree"). Following each section, the respondents could leave comments regarding the given item.

The preregistration indicated consensus on the given item if the interquartile range of its ratings was two or smaller. It defined support for an item if the median rating was six or higher (as in *Aczel et al., 2020*).

Each recommended practice found support and consensus from the 48 experts who completed ratings in our first round. For each item, the median rating was eight or higher with an interquartile range of two or lower. Thus, following our preregistration, there was no need to conduct additional consensus-survey rounds; all of the items were eligible to enter the guidance with consensual support. This high level of consensus might have been due to the experts' involvement in the preparatory round of the project. The summary table of the results is available at https://osf.io/qc7a8/.

### Finalizing the manuscript

The proposers drafted the manuscript and supplements. All texts and materials were sent to the expert panel members. Each contributor was encouraged to provide feedback on the manuscript, the report, and the suggested final version of the guidance. After all discussions, minor wording changes were implemented, as documented at https://osf.io/e39j4/. No contributor objected to the content and form of the submitted materials and all approved the final item list.

## Multi-analyst guidance

The final guidance includes ten recommended practices (*Table 1*) concerning the five main stages of multi-analyst studies. To further assist researchers in documenting multi-analyst projects, we also provide a modifiable reporting template (*Supplementary file 1*), as well as a reporting checklist (*Supplementary file 2*).

In addition to the Multi-analyst Guidance and Checklist, we provide practical considerations that can support the organization and execution of multi-analyst projects. This section contains various clarifications, recommendations, practical tools, and optional extensions, covering the five main stages of a multi-analyst project.

### Recruiting co-analysts
Choosing co-analysts
The term co-analyst refers to one researcher or team of researchers working together in a

multi-analyst project. Researchers can collaborate on the analyses, but if they do, we recommend that they submit the analyses as one co-analyst team, in order to ensure the independence of the analyses across teams. Researchers from the same lab or close collaborators should be able to submit separate reports in the multi-analyst project as long as they do not discuss their analyses with each other until the project rules allow that. The lead team may conduct an analysis themselves depending on the study goals and the design of the project (e.g., to set a performance baseline for comparing submitted models). Alternatively, the lead team may choose not to conduct an analysis themselves; in any case, they are expected to be transparent about their level of involvement as well as the timing (e.g., whether they conducted their analyses with or without knowing the results of the crowd of analysts).

Researchers should carefully consider both the breadth and depth of statistical and research-area expertise required for their project and should justify their choices about the required qualifications, skills, and credentials for analysts in the project. If the aim of the study is to explore what factors influence researchers' analytical choices, then it can be useful to seek "natural variation" (representativeness) within an expert community or to maximize diversity of the co-analysts along the dimensions where they might differ the most in their choices (e.g., experience, background, discipline, interest in the findings, intellectual allegiance to different theories, paradigmatic viewpoints).

Deciding on the number of co-analysts
To decide on the desired number of co-analysts, one has to consider which of the two main purposes of the multi-analyst method applies to the given project:

Checking the robustness of the conclusions
The aim here is solely to check whether different analysts obtain the same conclusions. Confidence in the stability of the conclusions decreases with divergent results and increases with convergent results. Many projects can achieve this aim by recruiting only one additional analyst, or a handful of further analysts. For example, the above-mentioned two analyses of the same dataset published in the journal *Surgery* (*Fields et al., 2019*; *Turner et al., 2019*) were sufficient to detect that the analytical space allows for opposite conclusions.

**Table 1.** Recommended practices for the main stages of the multi-analyst method.

| Stage | Recommended practices |
|---|---|
| Recruiting co-analysts | 1. Determine a minimum target number of co-analysts and outline clear eligibility criteria before recruiting co-analysts. We recommend that the final report justifies why these choices are adequate to achieve the study goals.<br>2. When recruiting co-analysts, inform them about (a) their tasks and responsibilities; (b) the project code of conduct (e.g., confidentiality/ non-disclosure agreements); (c) the plans for publishing the research report and presenting the data, analyses, and conclusion; (d) the conditions for an analysis to be included or excluded from the study; (e) whether their names will be publicly linked to the analyses; (f) the co-analysts' rights to update or revise their analyses; (g) the project time schedule; and (h) the nature and criteria of compensation (e.g., authorship). |
| Providing datasets, research questions, and research tasks | 3. Provide the datasets accompanied with a codebook that contains a comprehensive explanation of the variables and the datafile structure.<br>4. Ensure that co-analysts understand any restrictions on the use of the data, including issues of ethics, privacy, confidentiality, or ownership.<br>5. Provide the research questions (and potential theoretically derived hypotheses that should be tested) without communicating the lead team's preferred analysis choices or expectations about the conclusions. |
| Conducting the independent analyses | 6. To ensure independence, we recommend that co-analysts should not communicate with each other about their analyses until after all initial reports have been submitted. In general, it should be clearly explained why and at what stage co-analysts are allowed to communicate about the analyses (e.g., to detect errors or call attention to outlying data points). |
| Processing the results | 7. Require co-analysts to share with the lead team their results, the analysis code with explanatory comments (or a detailed description of their point-and-click analyses), their conclusions, and an explanation of how their conclusions follow from their results.<br>8. The lead team makes the commented code, results, and conclusions of all non-withdrawn analyses publicly available before or at the same time as submitting the research report. |
| Reporting the methods and results | 9. The lead team should report the multi-analyst process of the study, including (a) the justification for the number of co-analysts; (b) the eligibility criteria and recruitment of co-analysts; (c) how co-analysts were given the data sets and research questions; (d) how the independence of analyses was ensured; (e) the numbers of and reasons for withdrawals and omissions of analyses; (f) whether the lead team conducted an independent analysis; (g) how the results were processed; (h) the summary of the results of co-analysts; (i) and the limitations and potential biases of the study.<br>10. Data management should follow the FAIR principles (*Wilkinson et al., 2016*), and the research report should be transparent about access to the data and code for all analyses (*Aczel et al., 2020*). |

## Assessing the variability of the analyses

Those who wish to estimate the variability among the different analysis strategies often need to satisfy stricter demands. For example, studies that aim to assess how much the results vary among the analysts will require a larger number of co-analysts. When determining the number of co-analysts in such cases, the same factors need to be taken into consideration as in standard sample size estimation methods. For example, Botvinik-Nezer et al. (*Botvinik-Nezer et al., 2020a*) presented the analyses of 70 teams to demonstrate the divergence of results when analyzing a functional magnetic resonance imaging dataset.

## Recruiting co-analysts

Depending on the specific goal of the research, the recruitment of co-analysts can happen in several ways. Co-analysts can be recruited before or after obtaining the dataset. With stricter eligibility criteria, co-analysts can be invited individually from among topic experts or statistical experts. Follow-up open invitations can ask experts to suggest others to be invited. Alternatively, the lead team can open the opportunity to anyone to join the project as a co-analyst within the expert community (e.g., in professional society mailing lists and on social media), where expertise can be defined as the topic requires it.

It is important to note that whenever the co-authors' behavior is the subject of the study then they should be regarded similarly to human participants respecting ethical and data protection regulations. Useful templates for project advertisement and analyst surveys can be found in *Silberzahn et al., 2018*; *Schweinsberg et al., 2021*.

## Providing the dataset, research questions, and research tasks

### Providing the dataset

The lead team can invite the co-analysts to conduct data preprocessing (in addition to the main analysis). If the lead team decides to conduct the preprocessing themselves, showing their preprocessing methods can be informative to the co-analysts, but also has the potential to influence them if the preprocessing reflects some preference of methods or expectations of outcomes.

Before providing the dataset, the lead team should ensure that data management will comply

with legal (e.g., the General Data Protection Regulation (GDPR) in the European Union) and ethical regulations applying to all teams (see *Lundberg et al., 2019*). If the dataset contains personal information, a version should be provided where data can no longer be related to an individual. An alternative is to provide a simulated dataset and ask the co-analysts to provide code to analyze the data (*Drechsler, 2011*; *Quintana, 2020*). The lead team can then run the code on the actual data.

It is important that the co-analysts understand not just the available dataset but also any ancillary information that might affect their analyses (e.g., prior exclusion of outliers or handling of missing data in the blinded dataset). Providing a codebook that is accessible and understandable for researchers with different backgrounds is essential (*Kindel et al., 2019*).

## Providing the research question

The provided research question(s) should motivate the analysis conducted by the co-analysts. The research questions should be conveyed without specifying preferred analysis choices or expectations about the conclusions. Depending on the purpose of the project, the research questions can be more or less specific. While more specific research questions limit the analytical freedom of the co-analysts, less specific ones better explore the ways researchers can diverge in their operationalization of their question. A research question (e.g., "Is happiness age-dependent?") can be more specific when, for example, it is formulated as a directional hypothesis (e.g., "Are young people more happy than old ones?") or when the constructs are better operationalized (e.g., by defining what counts as young and happy).

## Providing the task

The multi-analyst approach can leave the operationalization of the research question to the co-analysts so that they can translate the theoretical question into the measurement. Taking this approach can reveal the operational variations of a question, but it can also make it difficult to compare the statistical results.

Requesting results in terms of standardized metrics (e.g., *t*-values, standardized beta, Cohen's *d*) makes it easier to compare results between co-analysts. The requested metric can be determined from the aim of the analysis (e.g., hypothesis testing, parameter estimation). It needs to be borne in mind, however, that this request

might bias the analysis strategies towards using methods that easily provide such a metric. A practical tool with instructions on reporting effect estimates can be found in *Parker et al., 2020*.

Co-analysts should be asked to keep a record of any code, derivatives etc. that were part of the analysis, at least until the manuscript is submitted and all relevant materials are (publicly) shared.

As an extension, the co-analysts can be asked to record considered but rejected analysis choices and the reasoning behind their choices (e.g., by commented code, log-books, or dedicated solutions such as DataExplained [*Schweinsberg et al., 2021*]). These logs can reflect where and why co-analysts diverge in their choices.

Robustness, or multiverse analyses (in the sense that each team is free to provide a series of outcomes instead of a single one) can also be part of the task of the co-analysts so that multiple analyses are conducted under alternative data analysis preprocessing choices.

## Communication with co-analysts

In projects with many co-analysts, keeping contact via a dedicated email address and automating some of the messages (e.g., automated emails when teams finished a stage in the process) can help streamline the communication and make the process less prone to human errors. For co-analyst teams with multiple members, it can be helpful for each team to nominate one member as the representative for communications.

If further information is provided to a co-analyst following specific questions, it can be useful to make sure the same information is provided to all teams, for example via a Q&A section of the project website, hosting weekly office hours where participants could ask questions, or via periodic email with updates.

## Conducting the independent analyses

### Preregistering the process and statistical analyses

We can distinguish *meta-* and *specific preregistrations*. Meta-preregistrations concern the plan of the whole multi-analyst project. It is good practice for the lead team to preregister how they would process, handle, and report the results of the co-analysts in order to prevent result-driven biases. This can be done in the form of a Registered Report at journals that invite such submissions (*Chambers, 2013*). Any meta-scientific questions, such as randomization of co-analysts to different conditions with variations in instructions or data, or covariates of interest

for studying associations to analytic variability, should be specified.

Specific preregistrations concern the analysis plans of the co-analysts. Requiring co-analysts to prepare a specific preregistration for each analysis can be a strategy to prevent overfitting and undisclosed flexibility. It makes sense to require it from either all or none of the teams in order to maintain equal treatment among them (unless the effect of preregistration is a focus of the study).

Requiring specific preregistrations may be misaligned with the goals of the project when the aim is to explore how the analytic choices are formed during the analyses, independent of initial plans. Under such circumstances, requiring specific preregistrations may be counterproductive. Nevertheless, the lead team can record their meta-preregistration that lays down the details of the multi-analyst project.

There are alternative solutions to prevent researchers from being biased by their data and results. For example, co-analysts could be provided with blinded datasets (*Dutilh et al., 2018*; *Starns et al., 2019*; *Gøtzsche, 1996*), simulated datasets (*Quintana, 2020*), or with a subset of the data (e.g., 11).

### *Processing the results*
#### Collecting the results
To facilitate summarizing the co-analysts' methods, results, and conclusions, the lead team can collect results through provided templates or survey forms that can structure analysts' reports. It is practical to ask the co-analysts at this stage to acknowledge that they did not communicate or cooperate with other co-analysts regarding the analysis in the project. It can also be helpful for the lead team if the co-analysts explain how their conclusions were derived from the results. In case preregistration was employed for any analyses, the template can also collect any deviations from the preregistered plan for inclusion in an online supplement.

To collect analytic code, it may be useful to require a container image (*Boettiger, 2015*; *Nüst et al., 2020*) or a portable version of the code that handles issues like software package availability (*Liu and Salganik, 2019*) (for a guideline see *Elmenreich et al., 2019*).

#### Validating the results
The lead team is recommended to ensure that each analyst's codes/procedures reproduce that analyst's submitted results. Computational reproducibility can be ascertained by running the code or repeating the analytic process by the lead team, but independent experts or the other co-analysts can also be invited to undertake this task (*Hurlin and Perignon, 2019*; *Pérignon et al., 2019*).

The project can leverage the crowd by asking co-analysts to review others' analyses, or the lead team can employ external statistical experts to assess analyses and detect major errors. The lead team can decide to omit analyses with major errors. In that case, the reasons for omission should be documented, and for transparency, the results of the omitted analyses should be included in an online supplement.

After all the analyses have been submitted and validated, the co-analysts could have the option in certain projects to inspect the work of the other analysts and freely withdraw their own analyses. This can be appropriate if seeing other analyses makes them aware of major mistakes or shortcomings in their analytic procedures. A potential bias in this process is that co-analysts might lose confidence in their analyses after seeing other, more senior, or more expert co-analysts' work. One way to decrease this potential bias is to follow a multi-stage process: after the first round of analyses is submitted, co-analysts could be allowed to see each other's analysis steps/code without knowing the identity of the co-analyst or the results of their analysis. It is the lead team's decision whether they allow co-analysts to correct or update their analyses after an external analyst or the co-analysts themselves find issues in their analyses.

Importantly, it is a minimum expectation that from the start of the project, the co-analysts should know about the conditions for their analyses to be included in, or omitted from, the study. All withdrawals, omissions, and updates of the results should be transparent in subsequent publications, for example in the supplementary materials.

### *Reporting the methods and results*
#### Recording contributorship
Using CRediT taxonomy can transparently record organizers' and co-analysts' contributions to the study. Practical tools (e.g., tenzing *Holcombe et al., 2020*) can make this task easier. Co-analysts can be invited to be co-authors and/or be compensated for their contribution in other ways (e.g., prizes, honorariums). Expectations for contribution and authorship should be communicated clearly at the outset.

Presenting the methods and results

Beyond a descriptive presentation of results in a table or graph, the reporting of the results of multi-analyst projects is not straightforward and remains an open area of research. Published reports of multi-analyst projects have adopted several effective methods for presenting results. For binary outcomes, Botvinik-Nezer et al. used a table with color coding (i.e., a binary heat map) to visualize outcomes across all teams (*Botvinik-Nezer et al., 2020b*). They overlaid each teams' confidence in their findings and added additional information about analytical paths in adjacent columns (*Supplementary file 1*, *Table 1*). For a project with a relatively small number of effect sizes for continuous outcomes, Schweinsberg et al. used interval plots combined with an indication of analytical choices underlying each estimate (*Schweinsberg et al., 2021*; Figure 3). Olsson Collentine et al. used funnel plots (Figure 2 in *Olsson-Collentine et al., 2020*), and Patel et al. used volcano plots to depict numerous, diverse outcomes with an intuitive depiction of clustering, akin to a multiverse analysis (Figures 1 and 2 in *Patel et al., 2015*).

If the main purpose is to estimate variability of analyses, it is interesting to investigate and report factors that might influence variability in the chosen analytic approaches and in the results obtained by these analytical approaches. If, on the other hand, the main purpose is to investigate the robustness of conclusions by assessing the degree to which different analysts obtain the same results, it is advisable to focus more on methods that produce only a single answer to the research question of interest. When each analysis team can provide multiple, distinct responses to the same research question, it becomes more difficult to explore how conclusions depend on the analysis choices because the individual analyses are no longer independent of each other.

The analytical approach of each co-analyst can be divided into discrete choices concerning, for instance, data preprocessing steps and decisions in model specification. If it is possible to recombine the individual choices (which will not always be the case as certain data preprocessing steps or method choices may only make sense if the aim is to fit a certain class of models), it may be worthwhile to create a larger set of possible analytical approaches that is made up of all possible combinations. In this case, the descriptive results of the multi-analyst project can be combined with a multiverse type approach (e.g., vibration of effects [*Patel et al., 2015*], multiverse analysis [*Steegen et al., 2016*], or specification curve [*Simonsohn et al., 2020*]) to quantify and compare the variability in results that can be explained by the different analytical choices (*Patel et al., 2015*; *Liu et al., 2021*). Additionally, this larger set of possible combinations can be helpful to present the results in an interactive user interface in which readers can explore how the results change as a function of certain analytical choices (*Liu et al., 2021*; *Dragicevic et al., 2019*). Finally, dividing the co-analysts' analytical approaches into individual choices may ultimately help in providing a unique answer to the research question of interest while accounting for the uncertainty in the choice of the analytical approach. While there are so far no approaches that would allow the derivation of a unique result that integrates all uncertain decisions, it may be a promising area of research to extend Bayesian approaches that account for model uncertainty (*Hoeting et al., 1999*) and measurement error (*Richardson and Gilks, 1993*).

To support the reporting of Multi-Analyst projects, we provide a freely modifiable *Reporting Template* available from here: https://osf.io/h9mgy/.

## Limitations

The present work does not cover all aspects of multi-analyst projects. For instance, the multi-analyst approach outlined here entails the independent analysis of one or more datasets, but it should be acknowledged that other crowd-sourced analysis approaches might not require such independence of the analyses. Some of our practical considerations reflect disagreement and/or uncertainty within our expert panel, so they remain underspecified. Those include how to determine the number or eligibility of co-analysts for a project, how best to assess the validity of each analysis; and how to measure robustness of conclusions. Therefore, we emphasize that this consensus-based guidance is a first step towards the broader adoption of the multi-analyst approach in empirical research, and we hope and expect that our recommendations will be developed further in response to user feedback. Users of this guidance can provide feedback and suggestions for revisions at https://forms.gle/2fVqZAD3KKHVUDKq7.

## Conclusions

This guidance document aims to facilitate adoption of the multi-analyst approach in both basic and clinical research. Although the multi-analyst

approach is at an incipient stage of adoption, we believe that the scientific benefits greatly outweigh the extra logistics required, especially for projects with high relevance for clinical practice and policy making. The approach should have particular relevance when it indicates that applying different analysis strategies to a given dataset may lead to conflicting results. The multi-analyst approach allows a systematic exploration of the analytical space to assess whether the reported results and conclusions are dependent on the chosen analysis strategy, ultimately improving the transparency, reliability, and credibility of research findings.

We hope that our guidance here and in guideline databases will make it easier for researchers to adopt this approach to empirical analyses. We encourage journals and funders to consider recommending or requesting independent analyses whenever it is crucial to know whether the conclusions are robust to alternative analysis strategies.

**Balazs Aczel** is at ELTE Eotvos Lorand University, Budapest, Hungary
aczel.balazs@ppk.elte.hu
http://orcid.org/0000-0001-9364-4988

**Barnabas Szaszi** is at ELTE Eotvos Lorand University, Budapest, Hungary
szaszi.barnabas@ppk.elte.hu
http://orcid.org/0000-0001-7078-2712

**Gustav Nilsonne** is at the Karolinska Institutet and Stockholm University, Stockholm, Sweden
http://orcid.org/0000-0001-5273-0150

**Olmo R van den Akker** is at Tilburg University, Tilburg, Netherlands

**Casper J Albers** is at the University of Groningen, Groningen, Netherlands

**Marcel ALM van Assen** is at Tilburg University, Tilburg, and Utrecht University, Utrecht, Netherlands

**Jojanneke A Bastiaansen** is at the University Medical Center Groningen, University of Groningen, Groningen, and Friesland Mental Health Care Services, Leeuwarden, Netherlands
http://orcid.org/0000-0003-4831-6402

**Daniel Benjamin** is at the University of California Los Angeles, Los Angeles, and the National Bureau of Economic Research, Cambridge, United States
http://orcid.org/0000-0002-2642-5416

**Udo Boehm** is at the University of Amsterdam, Amsterdam, Netherlands
http://orcid.org/0000-0002-8677-0721

**Rotem Botvinik-Nezer** is at Dartmouth College, Hanover, United State
http://orcid.org/0000-0003-2669-1877

**Laura F Bringmann** is at the University of Groningen, Groningen, Netherlands
http://orcid.org/0000-0002-8091-9935

**Niko A Busch** is at the University of Münster, Münster, Germany
http://orcid.org/0000-0003-4837-0345

**Emmanuel Caruyer** is at the University of Rennes, CNRS, Inria and Inserm, Rennes, France
http://orcid.org/0000-0002-8547-7726

**Andrea M Cataldo** is at McLean Hospital, Belmont, and Harvard Medical School, Boston, United States
http://orcid.org/0000-0003-2787-224X

**Nelson Cowan** is at the University of Missouri, Columbia, United States
http://orcid.org/0000-0003-3711-4338

**Andrew Delios** is at the National University of Singapore, Singapore
http://orcid.org/0000-0002-6791-227X

**Noah NN van Dongen** is at the University of Amsterdam, Amsterdam, Netherlands
http://orcid.org/0000-0003-0387-7388

**Chris Donkin** is at the University of New South Wales, Sydney, Australia

**Johnny B van Doorn** is at the University of Amsterdam, Amsterdam, Netherlands
http://orcid.org/0000-0003-0270-096X

**Anna Dreber** is at the Stockholm School of Economics, Stockholm, Sweden, and the University of Innsbruck, Innsbruck, Austria
http://orcid.org/0000-0003-3989-9941

**Gilles Dutilh** is at the University Hospital Basel, Basel, Switzerland

**Gary F Egan** is at Monash University, Melbourne, Australia
http://orcid.org/0000-0002-3186-4026

**Morton Ann Gernsbacher** is at the University of Wisconsin-Madison Madison, United States
http://orcid.org/0000-0003-0397-3329

**Rink Hoekstra** is at the University of Groningen, Groningen, Netherlands
http://orcid.org/0000-0002-1588-7527

**Sabine Hoffmann** is at Ludwig-Maximilians-University, Munich, Germany
http://orcid.org/0000-0001-6197-8801

**Felix Holzmeister** is at the University of Innsbruck, Innsbruck, Austria
http://orcid.org/0000-0001-9606-0427

**Juergen Huber** is at the University of Innsbruck, Innsbruck, Austria
http://orcid.org/0000-0003-0073-0321

**Magnus Johannesson** is at the Stockholm School of Economics, Stockholm, Sweden
http://orcid.org/0000-0001-8759-6393

**Kai J Jonas** is at Maastricht University, Maastricht, Netherlands

**Alexander T Kindel** is at Princeton University, Princeton, United States

**Michael Kirchler** is at the University of Innsbruck, Innsbruck, Austria
http://orcid.org/0000-0002-5416-2545

**Yoram K Kunkels** is at University Medical Center Groningen, University of Groningen, Groningen, Netherlands

D Stephen Lindsay is at the University of Victoria, Victoria, Canada

Jean-Francois Mangin is at Université Paris-Saclay, Paris, and Neurospin CEA, Paris, France
http://orcid.org/0000-0002-1612-461X

Dora Matzke is at Amsterdam University, Amsterdam, Netherlands

Marcus R Munafò is at the University of Bristol, Bristol, United Kingdom

Ben R Newell is at the University of New South Wales, Sydney, Australia
http://orcid.org/0000-0003-1898-205X

Brian A Nosek is at the Center for Open Science and the University of Virginia, Charlottesville, United States

Russell A Poldrack is at Stanford University, Stanford, United States
http://orcid.org/0000-0001-6755-0259

Don van Ravenzwaaij is at the University of Groningen, Groningen, Netherlands
http://orcid.org/0000-0002-5030-4091

Jörg Rieskamp is at the University of Basel, Basel, Switzerland
http://orcid.org/0000-0003-2632-8015

Matthew J Salganik is at Princeton University, Princeton, United States

Alexandra Sarafoglou is at Amsterdam University, Amsterdam, Netherlands

Tom Schonberg is at Tel Aviv University, Tel Aviv, Israel
http://orcid.org/0000-0002-4485-816X

Martin Schweinsberg is at ESMT Berlin, Berlin, Germany
http://orcid.org/0000-0003-3529-9463

David Shanks is at University College London, London, United Kingdom
http://orcid.org/0000-0002-4600-6323

Raphael Silberzahn is at the University of Sussex, Brighton, United Kingdom

Daniel J Simons is at the University of Illinois at Urbana-Champaign, Urbana-Champaign, United States

Barbara A Spellman is at the University of Virginia, Charlottesville, United States

Samuel St-Jean is at the University of Alberta, Edmonton, Canada, and Lund University, Lund, Sweden
http://orcid.org/0000-0002-8092-2974

Jeffrey J Starns is at the University of Massachusetts Amherst, Amherst, United States

Eric Luis Uhlmann is at INSEAD, Singapore, Singapore

Jelte Wicherts is at Tilburg University, Tilburg, Netherlands
http://orcid.org/0000-0003-2415-2933

Eric-Jan Wagenmakers is at Amsterdam University, Amsterdam, Netherlands

*Author contributions:* Balazs Aczel, Conceptualization, Methodology, Project administration, Writing – original draft, Writing – review and editing; Barnabas Szaszi, Conceptualization, Methodology, Writing – original draft, Writing – review and editing; Gustav Nilsonne, Conceptualization, Methodology, Writing – original draft, Writing – review and editing; Olmo R van den Akker, Writing – review and editing; Casper J Albers, Writing – review and editing; Marcel ALM van Assen, Writing – review and editing; Jojanneke A Bastiaansen, Writing – review and editing; Daniel Benjamin, Writing – review and editing; Udo Boehm, Writing – review and editing; Rotem Botvinik-Nezer, Writing – review and editing; Laura F Bringmann, Writing – review and editing; Niko A Busch, Writing – review and editing; Emmanuel Caruyer, Methodology, Writing – review and editing; Andrea M Cataldo, Writing – review and editing; Nelson Cowan, Writing – review and editing; Andrew Delios, Writing – review and editing; Noah NN van Dongen, Writing – review and editing; Chris Donkin, Writing – review and editing; Johnny B van Doorn, Writing – review and editing; Anna Dreber, Writing – review and editing; Gilles Dutilh, Writing – review and editing; Gary F Egan, Writing – review and editing; Morton Ann Gernsbacher, Writing – review and editing; Rink Hoekstra, Writing – review and editing; Sabine Hoffmann, Writing – review and editing; Felix Holzmeister, Writing – review and editing; Juergen Huber, Writing – review and editing; Magnus Johannesson, Writing – review and editing; Kai J Jonas, Writing – review and editing; Alexander T Kindel, Writing – review and editing; Michael Kirchler, Writing – review and editing; Yoram K Kunkels, Writing – review and editing; D Stephen Lindsay, Writing – review and editing; Jean-Francois Mangin, Writing – review and editing; Dora Matzke, Writing – review and editing; Marcus R Munafò, Writing – review and editing; Ben R Newell, Writing – review and editing; Brian A Nosek, Writing – review and editing; Russell A Poldrack, Writing – review and editing; Don van Ravenzwaaij, Writing – review and editing; Jörg Rieskamp, Writing – review and editing; Matthew J Salganik, Writing – review and editing; Alexandra Sarafoglou, Writing – review and editing; Tom Schonberg, Writing – review and editing; Martin Schweinsberg, Writing – review and editing; David Shanks, Writing – review and editing; Raphael Silberzahn, Writing – review and editing; Daniel J Simons, Writing – review and editing; Barbara A Spellman, Writing – review and editing; Samuel St-Jean, Writing – review and editing; Jeffrey J Starns, Writing – review and editing; Eric Luis Uhlmann, Writing – review and editing; Jelte Wicherts, Writing – review and editing; Eric-Jan Wagenmakers, Conceptualization, Methodology, Writing – original draft, Writing – review and editing

*Competing interests:* Brian A Nosek: Executive Director of the Center for Open Science, a non-profit technology and culture change organization with a mission to increase openness, integrity, and reproducibility of research. The other authors declare that no competing interests exist.

## Funding

| Funder | Grant reference number | Author |
|---|---|---|
| Netherlands Organisation for Scientific Research | 406-17-568 | Alexandra Sarafoglou |
| Natural Sciences and Engineering Research Council of Canada | BP-546283-2020 | Samuel St-Jean |
| Fonds de Recherche du Québec - Nature et Technologies | 290978 | Samuel St-Jean |
| European Research Council | 726361 | Jelte Wicherts Olmo R van den Akker |
| European Research Council | 681466 | Yoram K Kunkels |
| VIDI fellowship organisation | 016.Vidi.188.001 | Don van Ravenzwaaij |
| VENI fellowship grant | Veni 191G.037 | Laura F Bringmann |
| National Science Foundation | 1760052 | Matthew J Salganik |
| Weizmann Institute of Science | Israel National Postdoctoral Award Program for Advancing Women in Science | Rotem Botvinik-Nezer |
| John Templeton Foundation, Templeton World Charity Foundation, Templeton Religion Trust, and Arnold Ventures | | Brian A Nosek |
| Institut Européen d'Administration des Affaires | | Eric Luis Uhlmann |
| European Research Council | 640638 | Noah NN van Dongen |

The funders had no role in study design, data collection and interpretation, or the decision to submit the work for publication.

**Decision letter and Author response**
Decision letter https://doi.org/10.7554/eLife.72185.sa1
Author response https://doi.org/10.7554/eLife.72185.sa2

# Additional files

## Supplementary files
• Supplementary file 1. Reporting template for multi-analyst studies.

• Supplementary file 2. Reporting checklist for multi-analyst studies.

• Transparent reporting form

## Data availability
All anonymized data as well as the survey materials are publicly shared on the Open Science Framework page of the project: https://osf.io/4zvst/. Our methodology and data-analysis plan were preregistered. The preregistration document can be accessed at: https://osf.io/dgrua.

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
