## [Decision Letter]

**Decision letter after peer review:**

Thank you for submitting your article "Consensus-based guidance for conducting and reporting multi-analyst studies" to *eLife* for consideration as a Feature Article. Your article has been reviewed by three peer reviewers, and the evaluation has been overseen by the Peter Rogers, *eLife* Features Editor. The following individuals involved in review of your submission have agreed to reveal their identity: Florian Naudet; Ioana Cristea.

The reviewers and editors have discussed the reviews and we have drafted this decision letter to help you prepare a revised submission.

Summary:

The report describes a consensus-based guideline for multi-analyst studies. Overall, the manuscript provides very helpful suggestions, but I have some doubts about how much such a resource-intensive tool (beyond the rather simple version of independent reproducibility checks) will actually be used by researchers. As it stands now, the audience for such a guideline is both highly select and also I would imagine very limited (at least for now). On the other hand, as demonstrated by several papers the authors cite (such as Botvinik-Nezer et al.), the approach can have profound consequences for the practices of an entire field. However, there are a number of points that need to be addressed to make the article suitable for publication.

Essential revisions:

Section on Methods (for the expert consensus procedure)

1. Line 129: Please provide a reference for the "reactive Delphi" expert consensus procedure. Also, please comment on whether or not you followed any existing guidelines (eg CREDES) for reporting Delphi survey studies.

2. More information is needed on the process of recruiting experts. How was the list of existing many-analyst studies collected (e.g., systematic search, word of mouth)? What was the criteria for selecting "experts on research methodology"? What reasons were given by those experts who declined to get involved? Is there a minimal number of experts required for these kind of Delphi surveys? And what was the rationale for including only experts, and not a wider sample of people who might use the guideline in the future?

3. What was the rationale behind the threshold for expert consensus which was preregistered as IQR <= 2 and median >=6 (https://osf.io/dgrua)? I'm curious about why a minimum level of satisfaction wasn't sought (e.g. items 6 and 8 both have a response of "1", indicating maximum disagreement).

4. I think that more could be said in the text about the results of the consensus panel, and that consensus ratings for each item should at least be described in the main manuscript and not just by referencing a link to OSF.

5. It was a bit surprising to me that so there was so little disagreement between the expert panel on virtually all points of the guideline. This could be due to the fact that all the components of the guideline, with few exceptions, are pretty standard in terms of transparency and thus hard to disagree on (or might reflect that the relatively few multi-analyst studies conducted so far were conducted with an exceptionally high degree of rigor and transparency.) Please comment on this lack of disagreement.

Related to this, I found the section on "Practical Considerations" more nuanced and I assume there could have been some areas of disagreement in assembling this section. It would be interesting if the authors could reflect on some of these disagreements by, for instance, emphasizing points of discussion that were more controversial or where there was a more coagulated minority opinion, if any.

6. Sub-section on "Deciding on the number of co-analysts" (Lines 220-235).

My understanding is that the number of analysts needed may depend on the research question and also on the complexity of the dataset that would be analyzed. I would expect more discussion and guidance here. Currently, I don't think that this sub-section is really useful. Providing some examples would also be very helpful.

Section on Practical considerations

7. This section includes links/references for a couple of tools for any researchers who are considering embarking on their first many-analyst project (eg "instructions for reporting an effect estimate" and tenzing): it would be helpful to includes links/references for more such tools and resources, such as templates for providing the research task to many-analysts, a guide to automating emails, or a tutorial on using container images.

8. Line 271: Please provide examples of the standardized metrics that might be used for reporting results, and comment on how one might select one metric over another.

9. Please add a sub-section on "Providing the Research Questions" to the section "Providing the Dataset, Research Questions, and Research Tasks".

10. Sub-section on "Presenting the methods and results". Please give three or four examples of publications where the authors have done a good job of presenting their methods and results: please also cite the relevant figure(s) and/or table(s) for each example.

Other sections

11. Please add a paragraph to the "Conclusions" section on how you plan to disseminate these guidelines (eg, via the EQUATOR Network?)

12. There are guidelines on developing reporting guidelines for health research (eg Moher et al. 2010 Guidance for developers of health research reporting guidelines. DOI: https://doi.org/10.1371/journal.pmed.1000217). Please comment on whether or not you followed any such guidelines.

Improving the supplementary data

13. These comments are for the supplementary data file https://osf.io/qc7a8/ downloaded on 25 August 2021.

i) The excel formulas to calculate the Median and IQR are missing the rows for experts 45-49 (row 44 is blank, maybe this caused an error?). E.g. the formula in cell B53 is "=MEDIAN(B2:B44)", when it should be "=MEDIAN(B2:B50)". Fixing the formulas doesn't substantially change the results.

ii) There are no data descriptions provided in the excel file. It is not hard to work out what all the numbers and column headings mean by going back to the manuscript, but, given this is a conduct guideline and the last item is about the FAIR principles, it would be nice to lead by example and provide detailed metadata to make the document easy to re-use.

iii) Not all the data has been made publicly available. The registration page (https://osf.io/dgrua) said "All collected raw and processed anonymous data will be publicly shared on the OSF page of the project." Instead, only partial, processed data from the final consensus survey are provided (the free-text comments are not included). It would be great to see two raw files exported from Qualtrics (with names removed/anonymized): one for the preparatory survey, and one for the consensus survey.

---

## [Author Response]

Summary:The report describes a consensus-based guideline for multi-analyst studies. Overall, the manuscript provides very helpful suggestions, but I have some doubts about how much such a resource-intensive tool (beyond the rather simple version of independent reproducibility checks) will actually be used by researchers. As it stands now, the audience for such a guideline is both highly select and also I would imagine very limited (at least for now). On the other hand, as demonstrated by several papers the authors cite (such as Botvinik-Nezer et al.), the approach can have profound consequences for the practices of an entire field. However, there are a number of points that need to be addressed to make the article suitable for publication.

We appreciate these thoughts and believe that the scientific benefits greatly outweigh the extra logistics required, especially for projects with high relevance of important theoretical or policy making questions. The guidelines, checklist, and reporting template that we disseminate here were designed to decrease the burden of those who plan to conduct multi-analyst projects.

Essential revisions:Section on Methods (for the expert consensus procedure)1. Line 129: Please provide a reference for the "reactive Delphi" expert consensus procedure. Also, please comment on whether or not you followed any existing guidelines (eg CREDES) for reporting Delphi survey studies.

We reviewed the CREDES Guideline and now indicate that our approach meets those recommendations. We added a citation for this method on page 4. We also added clarification where needed (e.g., rationale behind using a Delphi-procedure, more information on the expert panel, discussion of how our approach affects potential biases).

2. More information is needed on the process of recruiting experts. What was the criteria for selecting "experts on research methodology"?

The contacted experts included all the organisers of known multi-analyst projects at the time as well as members of an expert panel of another methodological consensus project (Aczel, B., Szaszi, B., Sarafoglou, A., Kekecs, Z., Kucharský, Š., Benjamin, D.,.… and Wagenmakers, E. J. (2020). A consensus-based transparency checklist. Nature Human Behaviour, 4, 4-6.). We clarified it on page 5.

How was the list of existing many-analyst studies collected (e.g., systematic search, word of mouth)?

We conducted an unsystematic literature search and then asked those who had contributed to multi-analyst studies whether they knew of other examples in the literature. A social media call also allowed us to gather a range of multi-analyst studies.

https://mobile.twitter.com/BalazsAczel/status/1301801254348300288. We describe this approach on page 5.

What reasons were given by those experts who declined to get involved?

Three invited experts declined, noting their lack of insight in the topic. 28 invitees did not respond to our call. We now document these non-responses and declined invitations in the manuscript on page 5.

Is there a minimal number of experts required for these kind of Delphi surveys?

Murphy et al’s (1998) investigation of this question found that the reliability of group judgements increases substantially with every additional panel member up to 6 members. Beyond 12 members, the added benefits to reliability become minimal. But, a larger group can increase the diversity of perspectives and makes the result somewhat more robust. We had 50 members in our expert panel.

Murphy, M. K., Black, N. A., Lamping, D. L., McKee, C. M., Sanderson, C. F., Askham, J., and Marteau, T. (1998). Consensus development methods, and their use in clinical guideline development. Health technology assessment (Winchester, England), 2(3), i–88.

And what was the rationale for including only experts, and not a wider sample of people who might use the guideline in the future?

As the multi-analyst approach concerns specific methodological issues, we sought input primarily from researchers with relevant expertise and experience. In the manuscript, we emphasize that this guidance is a first step toward the broader adoption of the multi-analyst approach. We hope that our recommendations will be developed further based on feedback from adopters. We added a survey link to page 14 to collect such feedback and suggestions.

3. What was the rationale behind the threshold for expert consensus which was preregistered as IQR <= 2 and median >=6 (https://osf.io/dgrua)? I'm curious about why a minimum level of satisfaction wasn't sought (e.g. items 6 and 8 both have a response of "1", indicating maximum disagreement).

We used those thresholds based on this consensus paper:

https://www.nature.com/articles/s41562-019-0772-6. We judged that coherence (IQR) and summary level of support (median) are better indicators of the panel’s general thinking than are outlier values. We now cite this reference on page 5.

4. I think that more could be said in the text about the results of the consensus panel, and that consensus ratings for each item should at least be described in the main manuscript and not just by referencing a link to OSF.

We have improved the prose in that section. The paragraphs now are in page 5-6:

“The preregistration indicated consensus on the given item if the interquartile range of its ratings was 2 or smaller. […] The summary table of the results is available at https://osf.io/qc7a8/”.

As we had relatively high agreement with low variance for each item, we would prefer not to include information on each item in the paper. If the editor feels it’s essential, we could add a table (such as the one below) to the manuscript, but we felt that it would provide limited additional information beyond the description in the text.

**Author response table 1. sa2table1:** 

	Item1	Item2	Item3	Item4	Item5	Item6	Item7	Item8	Item9	Item10
Median ratings	8	9	9	9	9	9	8.5	9	9	9
Interquar-tile range	2	1	1	0	1.25	1.25	1.25	1	1	

5. It was a bit surprising to me that so there was so little disagreement between the expert panel on virtually all points of the guideline. This could be due to the fact that all the components of the guideline, with few exceptions, are pretty standard in terms of transparency and thus hard to disagree on (or might reflect that the relatively few multi-analyst studies conducted so far were conducted with an exceptionally high degree of rigor and transparency.) Please comment on this lack of disagreement.

The high level of agreement might be due to the fact that we had a Preparatory Round before the consensus ratings. During this round, the experts could comment on each item and the overall content of the guidance. Based on their feedback, the core team updated and revised the initial draft. Then, the experts had a new option to edit and comment on the content. As a result, the item list that they rated had already incorporated many of their insights. We now discuss this procedure more fully and note that it could contribute to the high levels of agreement. (See quote in previous reply.)

Related to this, I found the section on "Practical Considerations" more nuanced and I assume there could have been some areas of disagreement in assembling this section. It would be interesting if the authors could reflect on some of these disagreements by, for instance, emphasizing points of discussion that were more controversial or where there was a more coagulated minority opinion, if any.

Thank you for this suggestion. We have added discussion of these disagreements to our Limitations section on page 13-14:

“Some of our practical considerations reflect disagreement and/or uncertainty within our expert panel, so they remain underspecified. […] Therefore, we emphasise that this consensus-based guidance is a first step towards the broader adoption of the multi-analyst approach in empirical research, and we hope and expect that our recommendations will be developed further in response to user feedback.”

6. Sub-section on "Deciding on the number of co-analysts" (Lines 220-235).My understanding is that the number of analysts needed may depend on the research question and also on the complexity of the dataset that would be analyzed. I would expect more discussion and guidance here. Currently, I don't think that this sub-section is really useful. Providing some examples would also be very helpful.

We see value in drawing attention to the two distinct motivations for employing a multi-analyst approach – they have different requirements. We have added examples to each subsection on page 8.

Section on Practical considerations7. This section includes links/references for a couple of tools for any researchers who are considering embarking on their first many-analyst project (eg "instructions for reporting an effect estimate" and tenzing): it would be helpful to includes links/references for more such tools and resources, such as templates for providing the research task to many-analysts, a guide to automating emails, or a tutorial on using container images.

We added references to further templates and guidelines on pages 9 and 11.

8. Line 271: Please provide examples of the standardized metrics that might be used for reporting results, and comment on how one might select one metric over another.

We added examples and explanations on page 10.

9. Please add a sub-section on "Providing the Research Questions" to the section "Providing the Dataset, Research Questions, and Research Tasks".

Done.

10. Sub-section on "Presenting the methods and results". Please give three or four examples of publications where the authors have done a good job of presenting their methods and results: please also cite the relevant figure(s) and/or table(s) for each example.

We badded the following paragraph to this subsection with examples and relevant figure numbers on pages 12-13:

“Published reports of multi-analyst projects have adopted several effective methods for presenting results For binary outcomes, Botvinik-Nezer et al. (39) used a table with colour coding (i.e., a binary heat map) to visualise outcomes across all teams. […] Olsson Collentine et al. (40) (Figure 2) used funnel plots and Patel et al. (7) (Figures 1 and 2) used volcano plots to depict numerous, diverse outcomes with an intuitive depiction of clustering (akin to a multiverse analysis).”

Other sections11. Please add a paragraph to the "Conclusions" section on how you plan to disseminate these guidelines (eg, via the EQUATOR Network?)

Added it on page 14.

12. There are guidelines on developing reporting guidelines for health research (eg Moher et al. 2010 Guidance for developers of health research reporting guidelines. DOI: https://doi.org/10.1371/journal.pmed.1000217). Please comment on whether or not you followed any such guidelines.

We reviewed CREDES, a guideline for conducting and reporting Delphi studies, and added the following sentence to the manuscript on page 4:

“The applied consensus procedure and its reporting satisfy the recommendations of the CREDES (21) guideline on conducting and reporting Delphi studies.”

Improving the supplementary data13. These comments are for the supplementary data file https://osf.io/qc7a8/ downloaded on 25 August 2021.i) The excel formulas to calculate the Median and IQR are missing the rows for experts 45-49 (row 44 is blank, maybe this caused an error?). E.g. the formula in cell B53 is "=MEDIAN(B2:B44)", when it should be "=MEDIAN(B2:B50)". Fixing the formulas doesn't substantially change the results.

We really appreciate your thorough review of the accompanying code! Because of the blank row, our calculation included a mistake. We fixed the formulas and updated the supplementary materials and main text accordingly. Fixing the formulas does not substantially change the results.

ii) There are no data descriptions provided in the excel file. It is not hard to work out what all the numbers and column headings mean by going back to the manuscript, but, given this is a conduct guideline and the last item is about the FAIR principles, it would be nice to lead by example and provide detailed metadata to make the document easy to re-use.

We added detailed meta-data to the data file to make it easier to use.

iii) Not all the data has been made publicly available. The registration page (https://osf.io/dgrua) said "All collected raw and processed anonymous data will be publicly shared on the OSF page of the project." Instead, only partial, processed data from the final consensus survey are provided (the free-text comments are not included). It would be great to see two raw files exported from Qualtrics (with names removed/anonymized): one for the preparatory survey, and one for the consensus survey.

We have added the anonymized raw files to the OSF page.